

# Ginsenoside Rb1, salvianolic acid B and their combination modulate gut microbiota and improve glucolipid metabolism in high-fat diet induced obese mice

Ying Bai[1,*], Xueli Bao[2,*], Qianqian Mu[3], Xin Fang[2], Ruyuan Zhu[1], Chenyue Liu[1], Fangfang Mo[1], Dongwei Zhang[1], Guangjian Jiang[1], Ping Li[2], Sihua Gao[1] and Dandan Zhao[1]

[1] College of Traditional Chinese Medicine, Beijing University of Chinese Medicine, Beijing, China
[2] Third Affiliated Hospital, Beijing University of Chinese Medicine, Beijing, China
[3] Dongzhimen Hospital, Beijing University of Chinese Medicine, Beijing, China
[*] These authors contributed equally to this work.

Corresponding authors
Sihua Gao, gaosihua1216@163.com
Dandan Zhao, tcmzdd@sina.com

## ABSTRACT

**Background.** To observe the effect of ginsenoside Rb1, salvianolic acid B and their combination on glucolipid metabolism and structural changes of gut microbiota.
**Methods.** Eight-week-old C57BL/6J mice were fed 45% high-fat diet to induce obesity. The obese mice were randomly divided into four groups, Con group as model control, ginsenoside Rb1 (Rb1) group, salvianolic acid B (SalB) group and ginsenoside Rb1+ salvianolic acid B (Rb1SalB) group. Mice in Rb1, SalB and Rb1SalB group were treated by gavage with ginsenoside Rb1, salvianolic acid B and the combination of the two ingredients, respectively. While mice in Con group were given the same amount of sterile water. The intervention lasted 8 weeks. Body weight and fasting blood glucose were measured every 2 weeks. Oral glucose tolerance test was conducted on the 4th and 8th week of drug intervention. At the end of the experiment, total cholesterol, triglyceride, high density lipoprotein cholesterol, low density lipoprotein cholesterol and non-esterified fatty acid content as well as glycated hemoglobin were measured and feces were collected for 16S rDNA sequencing.
**Results.** Both ginsenoside Rb1 and Rb1SalB combination decreased body weight significantly ($P < 0.05$). Ginsenoside Rb1, salvianolic acid B and their combination alleviated fasting blood glucose, glycated hemoglobin and blood lipid profiles effectively ($P < 0.05$, compared with the corresponding indicators in Con group). Oral glucose tolerance test results at the 8th week showed that glucose tolerance was significantly improved in all three treatment groups. Ginsenoside Rb1, salvianolic acid B and their combination reduced the overall diversity of gut microbiota in feces and changed the microbial composition of the obese mice. LDA effect size (LefSe) analysis revealed the key indicator taxa corresponding to the treatment.
**Conclusion.** Ginsenoside Rb1, salvianolic acid B and their combination could lower blood glucose and lipid level, and improve glucose tolerance of obese mice. The above effect may be at least partially through modulation of gut microbial composition.

## BACKGROUND

We are now in an era with a widespread occurrence of obesity. According to the WHO report, in 2016 over 650 million adults were considered to be obese (BMI $\geq$ 30 kg/m$^2$) (*World Health Organization (WHO), 2016*). The imbalance between excessive fuel surfeit (calorie consumption) and lack of energy expenditure (physical exercises) contributes to the rapid growth of obesity, which is the trigger of a series of metabolic disorders such as type 2 diabetes mellitus (*Singh et al., 2015*). Glucose and lipid are the most basic forms of energy, so the glucolipid metabolic disorder is essential for development of obesity. Considering the fact that obesity accounts for enormous healthcare expenditure and leads to various short- and long-term medical consequences, such as hyperlipidemia, diabetes, stroke, coronary heart disease and so on, it is critical to investigate the solution to control this global concern (*Flegal et al., 2007*; *Withrow & Alter, 2011*).

In the last decade, people gained increased understanding about human gut microbiota as a critical player in health and diseases. Having co-evolved with humans, gut microbiota makes a significant contribution to human biology and development (*Li et al., 2016*). Albeit gut microbiota comprises a small percentage of human body weight, it is influenced by health condition, diet, medication, environment, and so on. Accumulating evidence indicates that gut microbiota dysbiosis contributes to the onset of obesity via affecting host energy metabolism and interfering with immune system (*Barlow, Yu & Mathur, 2015*; *Jiao et al., 2019*; *Lee, Sears & Maruthur, 2019*; *Qin et al., 2012*). On one hand, high fat diet could induce abnormal composition of microbiota and affect the host homeostasis; on the other hand, changes in the gut microbiota could be seen as early signal of some metabolic disorders. In this way, modulation of gut microbiota structure and adjustment of their metabolism appear to be a novel approach and a potential target of treating obesity and diabetes. Many drugs, active ingredients from herbal medicine and foods have been shown to play an anti-diabetic role by regulating the relative proportion of gut microbiota, and exert beneficial effect on microbial metabolites like short chain fatty acids (SCFAs), bile acids and so on (*Garcia-Mazcorro et al., 2018*; *Jiao et al., 2019*; *Zhang et al., 2012*).

Ginsenoside Rb1 and salvianolic acid B are two active components extracted from *Panax ginseng C.A.Mey.* and *Salvia miltiorrhiza Bunge*, respectively. They are commonly used for controlling of cardiovascular, endocrine and immunological diseases (*Jia et al., 2008*; *Shi et al., 2019*) Our previous study has confirmed that ginsenoside Rb1 could decrease body weight, ameliorate glucolipid metabolism, enhance skeletal muscle endurance and increase insulin sensitivity by activating AMPK signaling pathway in obese mice (*Zhao et al., 2019a*). Other pharmaceutical mechanisms of ginsenoside Rb1 include amelioration of liver fat accumulation, promotion of adipocytes browning via modulating PPARγ signaling, and regulation of several gut peptide and hypothalamic feeding center function (*Liu et al., 2013*; *Mu et al., 2015*; *Shen et al., 2015*; *Xiong et al., 2010*; *Yu et al., 2015*). As reported,

salvianolic acid B could also improve glucose tolerance and insulin sensitivity by activating AMPK signaling in skeletal and hepatic tissues, attenuating oxidative stress and improving mitochondrial function, and regulating lipogenesis and adipocyte differentiation (*An et al., 2019*; *Huang et al., 2015*; *Huang et al., 2016*; *Raoufi et al., 2015*; *Tao et al., 2017*; *Wang et al., 2019a*; *Wang et al., 2019b*; *Pan et al., 2018*; *Zhai et al., 2019*). Besides, studies have shown that metabolism of ginsenoside Rb1 into compound K might be dependent on specific gut microbiota composition (*Kim et al., 2013*). Also, salvianolic acid B could protect from DSS-induced colitis and diet induced obesity by altering gut microbiota composition (*Li et al., 2020*; *Wu et al., 2018*). These evidence all suggest the close interactions between gut microbiota and the two ingredients. Therefore, we hypothesized that ginsenoside Rb1 and salvianolic acid B and their combination might protect from obesity related weight gain and glucolipid disorder through alteration of gut microbiota composition. To validate this, we performed the current study.

## MATERIALS & METHODS

### Animal and feed

Eight-week-old male C57BL/6J mice were purchased from Beijing Sibeifu Biotechnology Co., Ltd. (Beijing, 2014-0004). Animal experiment in this study were performed in the barrier environment laboratory of Beijing University of Chinese Medicine. Mice were housed in SPF level animal facilities with tap water and food ad libitum. There were 4 mice per cage. Both water and feed were sterilized before use. The light and dark cycle was 12/12 h and the room temperature was controlled at $23 \pm 2\,°C$ with $55 \pm 10\%$ relative humidity. Both high-fat diet (MD12032, 45% energy from fat) and standard chow diet (MD12031, 10% energy from fat) was obtained from Jiangsu Medicine Biomedical Co. Ltd. (Jiangsu, China). All the protocols were carried out in accordance with Guide for the Care and Use of Laboratory Animals (Beijing University of Chinese Medicine) strictly. The Animal Ethics Committee of Beijing University of Chinese Medicine provided full approval for this study (NO. BUCM-4-2016061701-3001).

### Experimental drug

Ginsenoside Rb1 and salvianolic acid B were purchased from Chengdu Puruifa Technology Development Co., Ltd. (Chengdu, China) and stored in refrigerator at $4\,°C$. The required concentration of suspension was prepared with ultrapure water before gavage.

### Reagents and equipment

Hitachi 7080 automatic biochemical analyzer (Tokyo, Japan) was used to detect blood biochemical indexes. glycated hemoglobin (HbA1c, A056-1-1), total triglyceride (TG, A110-1-1), total cholesterol (TC, A111-1-1), high density lipoprotein-cholesterol (HDL-c, A112-1-1), low density lipoprotein-cholesterol (LDL-c, A113-1-1) and non-esterified fatty acid (NEFA, A042-2-1) kits were purchased from Nanjing Jiancheng Bioengineering Institute (Nanjing, China).

## Animal experiment

The experimental design of this study was shown in Fig. 1A. Male C57BL/6J mice aged 8 weeks were fed with 45% HFD for 12 weeks after one-week of adaptive feeding with standard chow. After 12 weeks, body weight (BW) was measured. Mice weighed 20% heavier than the same age mice fed standard chow diet were taken as the diet induced obese (DIO) mice. The normal level BW of C57BL6/J mice was obtained via the Jackson Laboratory website (https://www.jax.org/jax-mice-and-services/strain-data-sheet-pages/body-weight-chart-000664#). The DIO mice were then divided into model control group (Con group), ginsenoside Rb1 group (Rb1 group), salvianolic acid B group (SalB group), ginsenoside Rb1 + salvianolic acid B group (Rb1SalB group) randomly (8 mice in each group). Mice were given different kinds of treatment by gavage for 8 consecutive weeks. Rb1 group: ginsenoside Rb1 20 mg/kg d, SalB group: salvianolic acid B 100 mg/kg d, Rb1SalB group: ginsenoside Rb1 20 mg/kg d + salvianolic acid B 100 mg/kg d. Mice in Con group were given the same amount of sterile water. At the end of the experiment, all mice were fasted overnight and anesthetized by 1% sodium pentobarbital next morning. Then blood samples were taken by extracting the eyeballs. Finally, mice were executed by breaking their cervical vertebra. Criteria established for euthanizing animals prior to the planned end of the experiment is as follow. Euthanasia should be given to the dying or sick mice and those whose body weight did not meet the required standard after HFD feeding. When applying this, try to reduce the pain of animals, try to avoid their panic and struggle, shorten the death time as far as possible, and pay attention to the safety of laboratory personnel. Cervical dislocation was used for euthanizing mice.

## Detected indicators and methods

During the experiment, blood glucose levels and body weight were measured every 2 weeks. Oral glucose tolerance test (OGTT) was conducted at 4th and 8th week: after fasting overnight, mice were given 2.0 g/kg body weight of glucose by gavage, and the blood glucose was measured before (0 min) and 30, 60, 90 and 120 min after gavage. At the end of the experiment, blood samples were collected and centrifuged to obtain the serum. The levels of TG, TC, HDL-c, LDL-c, NEFA and HbA1c were determined according to the instructions of the corresponding kits.

## Fecal DNA extraction and 16S rDNA high-throughput sequencing

Twenty-four hours after the last treatment, the feces of mice in each group were collected, placed in dry sterile centrifugal tubes, frozen in liquid nitrogen, and then sent to Coyote Medical Laboratory (Beijing, China) for high-throughput 16S rDNA sequencing. DNA was extracted from the collected samples following the instruction with specific TIANamp stool DNA kit (TIANGEN Biotech, Beijing, China). The V3-V4 region the 16S rRNA was amplified using primers 341f (5′-CCTAYGGGRBGCASCAG-3′) and 806r (5′-GGACTACNNGGGTATCTAAT-3′) under the condition of an initial 3-min enzyme activation step at 95 °C; 20 cycles of 15 s at 98 °C, 30 s at 50 °C, and 40 s at 72 °C; and 10 min at 72 °C (*Klindworth, Pruesse & Schweer, 2013*). The PCR recipes contain 333 nmol of forward and reverse primers, 25 ng of input DNA and KAPA HiFi PCR Master Mix (Kapa

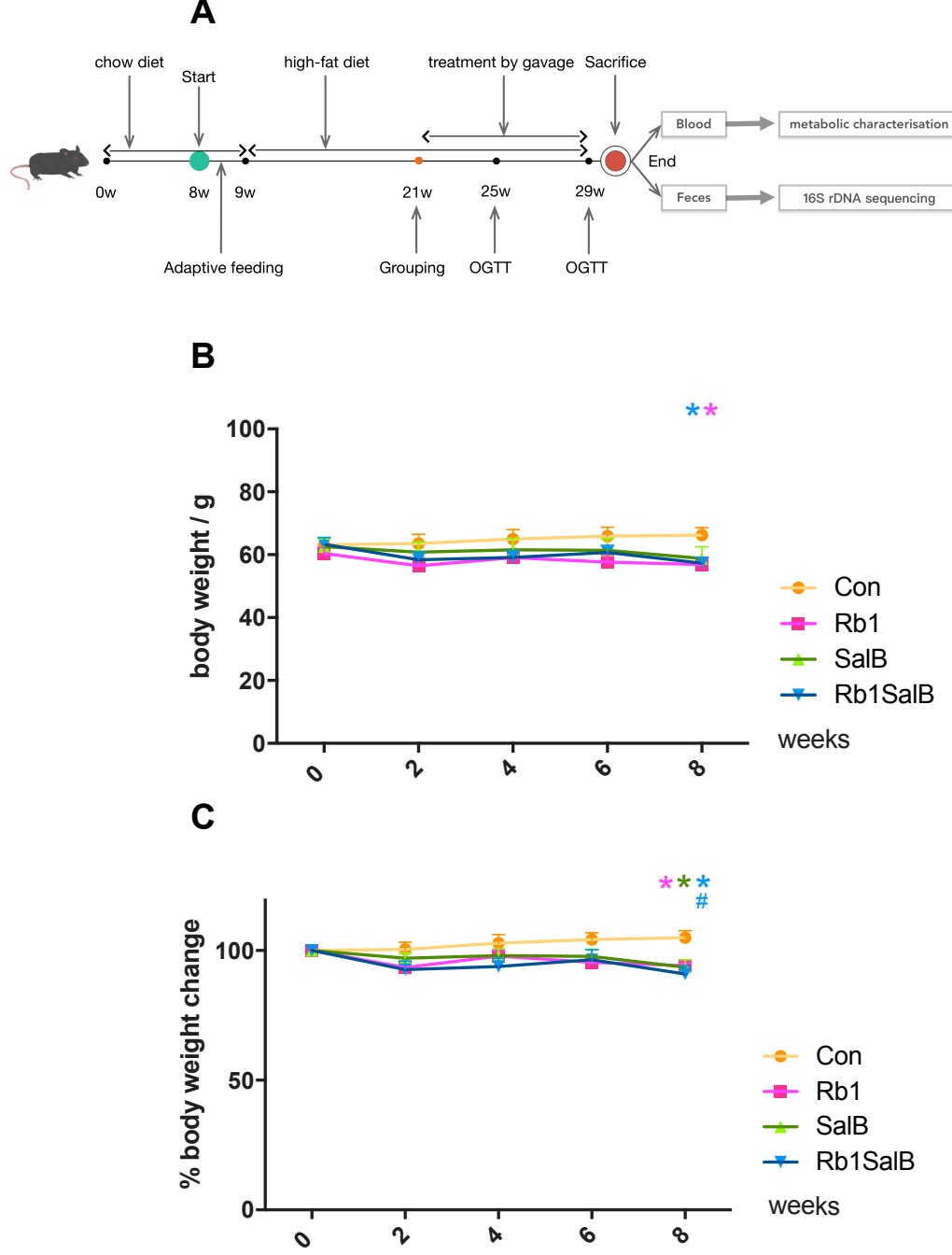

**Figure 1 Effects of Ginsenoside Rb1, Salvianolic acid B and their combination on body weight and weight gain in HFD-induced obese mice.** (A) Experimental design of present study. (B) Body weight in different groups. (C) Percentage of body weight change with the timeline in different groups. The ordinate is the percentage of body weight change compared with body weight at 0 week. Eight mice in each group. Differences were assessed by ANOVA. * Compared with Con group, $P < 0.05$; # Compared with original body weight (0 week), $P < 0.05$.

Biosystems, Boston, MA, USA). Next, the amplification products were purified applying VAHTS DNA clean beads (Vazyme Biotech Co., Ltd., Nanjing, China). The library was prepared and undergone quality control (*Wu et al., 2019*). Finally, the library was checked by electrophoresis through a 1% agarose gel and compared to a molecular weight standard (100 bp) and sequenced on Illumina HiSeq 2500 (PE250) sequencing platform using paired-end strategy. Raw tags were obtained from the original data through sequence assembly, then base sequence quality filtering and chimera filtering were carried out to obtain effective tags for analysis using the GOLD database and the UCHIME algorithm. According to 97% sequence consistency (*Quast et al., 2013*), the effective tags were clustered to get Operational Taxonomic Units (OTUs). Chimera detection and removal were assessed using GOLD database and UCHIME algorithm. Then the representative sequences of OTUs were compared with the 16S rDNA Greengenes reference database to get the results of taxa annotation (*Wu et al., 2019*).

Sequencing analysis includes OTUs taxa annotation, alpha diversity analysis, beta diversity analysis and significantly different taxa analysis. OTUs taxa annotation: the top 10 taxa in each taxonomic level of each group were plotted as bar plots to visualize the taxa with higher relative abundance and their proportion at different taxonomic levels. Venn diagrams were drawn to reflect the number of OTUs shared between groups and unique within each group. Alpha diversity analysis was the analysis of taxa diversity within a single sample. First, the rarefaction curves were drawn, and then the alpha diversity index was calculated by QIIME software (*Caporaso, Kuczynski & Stombaugh, 2010*). In this study, Chao 1 index and Shannon index were used to evaluate the microbiota richness and evenness of a single sample. Beta diversity analysis was conducted using non-metric multi-dimensional scaling analysis based on weighted and unweighted unifrac distance matrix. Beta diversity analysis was carried out to compare the differences in microbial composition among groups (*Lozupone et al., 2011*). LDA effect size (LEfSe) difference analysis was used to identify taxa with significant differences (*Segata et al., 2011*). The raw sequence data is available in NCBI Sequence Read Archive (SRA) under the accession PRJNA610166.

## Statistical analysis

SPSS 20.0 statistical software was used for data analysis. Values were expressed as means $\pm$ standard error. One-way ANOVA was used for multiple group comparison and Fisher's LSD method was used for comparison between two groups. $P < 0.05$ was considered statistically significant. GraphPad Prism7 was used for data management and plotting.

## RESULTS

### Effect of Rb1, SalB and their combination on body weight in DIO mice

The body weight of mice in each group was basically at the same level before drug intervention ($P > 0.05$, Fig. 1B). After 8 weeks of treatment, weight of mice in Rb1, SalB and Rb1SalB group decreased, while the weight of mice in the Con group increased. At the 8th week, mice in Rb1, Rb1SalB group weighed significantly lower than the Con group

($P < 0.05$, Fig. 1B). Overall, mice in the Con group continued to put on weight (5.00% of 0 week BW), while mice in Rb1 group lost 5.87%, in SalB group 6.45%, and in Rb1SalB group 9.00% of body weight. The difference between the three treatment groups and the Con group was statistically significant ($P < 0.05$, Fig. 1C). Our previous study showed that both Rb1 and SalB treatment exhibited no effect on food intake ($P < 0.05$, Fig. S1). Taken together, we conclude that Rb1, SalB and their combination might prevent high-fat diet induced weight gain without affecting food intake.

## Effect of Rb1, SalB and their combination on blood glucose of DIO mice

After intervention with Rb1, SalB or both, FBG levels presented a decreasing trend, and this hypoglycemic effect became more effective with the prolong of treatment time ($P < 0.05$, Fig. 2A). The levels of HbA1c in Rb1 group (6.55 ± 1.09%), SalB group (6.73 ± 0.52%) and Rb1SalB group (6.45 ± 0.13%) were significantly lower than that in the Con group (8.02 ± 1.19%, $P < 0.05$, Fig. 2B). To show the changes of glucose metabolism in mice clearly, OGTT was conducted at 4th and 8th week during drug administration (Figs. 2C–2F). The results suggested that mice in Rb1, SalB and Rb1SalB group showed prominently improved glucose tolerance than their littermates in Con group ($P < 0.05$), and this effect was more remarkable at 8th week than 4th week (Figs. 2C and 2E). Furthermore, the area under curve (AUC) of blood glucose was 50.48 ± 4.54 in Con group, 38.85 ± 1.61 in Rb1 group, 35.04 ± 3.03 in SalB group and 41.80 ± 3.92 in Rb1SalB group (Fig. 2D). And the AUC in the Con group was still the largest (AUC 45.13 ± 2.42), followed by Rb1 group (34.25 ± 2.45), SalB group (34.54 ± 2.05) and Rb1SalB group (32.44 ± 2.27) at 8th week (Fig. 2F). This indicated that Rb1, SalB and the combination of the two agents could effectively decrease the fasting blood glucose level, and improve glucose tolerance of DIO mice.

## Effect of Rb1, SalB and their combination on blood lipid content in DIO mice

Next, we measured the blood lipid content among four groups of mice. Compared with the Con group, TC, TG and LDL-c content of mice in Rb1 group were much lower; and TC, TG, LDL-c, NEFA content of mice in SalB and Rb1SalB group decreased significantly ($P < 0.05$, Figs. 3A–3C, 3E). However, HDL-c content among all groups showed no significant difference ($P < 0.05$, Fig. 3D). These results suggest that Rb1, SalB and Rb1SalB could reduce blood lipid accumulation caused by long time HFD feeding.

### Gut microbiota sequencing analysis in feces
#### Statistics of sequencing results
A total of 4,002,927 reads and 3207 OTUs were obtained from 32 samples out of four groups (Table 1). The number of reads obtained from Rb1 group was the highest (146063.6 ± 24658.80), while that in Rb1SalB group was the lowest (108817.4 ± 54935.34). All sequence datasets were rarefied to the smallest number of reads before further comparison. Con group had the most OTUs (692.50 ± 174.47). After the treatment of Rb1 and SalB, the number of OTUs decreased (OTU 627.88 ± 78.78 in the Rb1 group and 636.00 ± 178.83 in SalB group), while the Rb1SalB group exhibited the least number of OTUs (531.50 ± 78.46

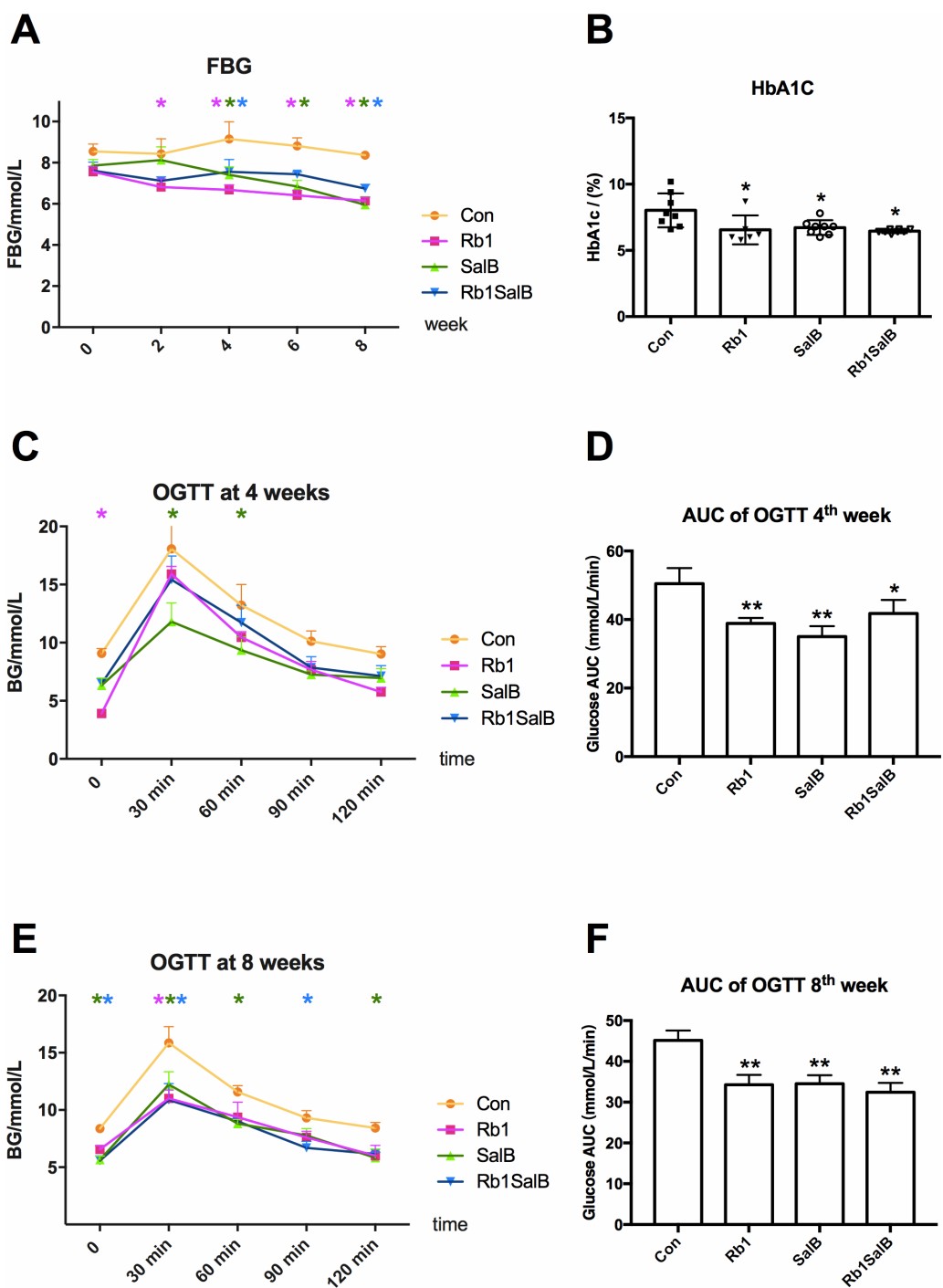

**Figure 2 Effects of Ginsenoside Rb1, Salvianolic acid B and their combination on blood glucose.** (A) Fasting blood glucose levels in HFD-induced obese mice of different groups. FBG was measured every 2 weeks. (B) HbA1c content of mice in different groups. Eight mice in each group. (C, E) Oral glucose tolerance test was carried out at the 4th, 8 th week. (D, F) Area under curve (AUC) of OGTT at the 4th and 8th week, respectively. For OGTT, 4 mice in each group. * Compared with Con group, $P < 0.05$; ** compared with Con group, $P < 0.01$.

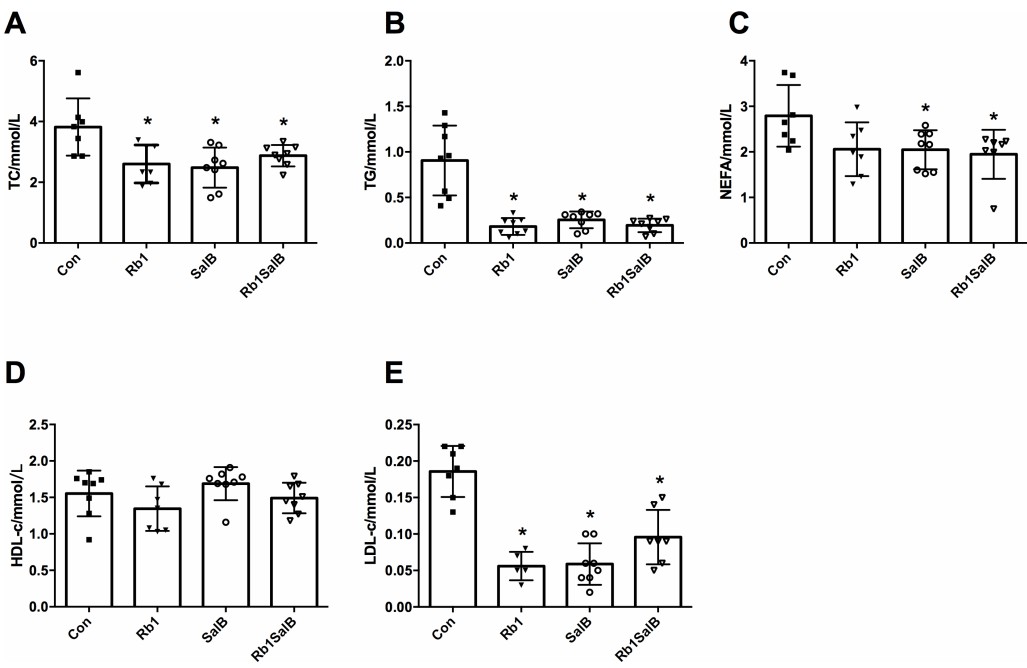

**Figure 3** **Effects of Ginsenoside Rb1, Salvianolic acid B and their combination on serum blood lipid content.** (A) Total cholesterol (TC), (B) triglyceride (TG), (C) non-esterified fatty acid (NEFA), (D) high density lipoprotein cholesterol (HDL-c), (E) low density lipoprotein cholesterol (LDL-c) were measured after 8 weeks' intervention. Eight mice in each group. * $P < 0.05$ compared with Con group.

**Table 1** **Effective reads and OTUs in four groups.**

| Group name | Effective reads | OTU |
|---|---|---|
| Con | 129083.50 ± 32821.23 | 692.50 ±174.47 |
| Rb1 | 146063.63 ± 24658.80 | 627.88 ±78.78 |
| SalB | 116401.38 ± 44628.07 | 636.00 ±178.83 |
| Rb1SalB | 108817.38 ± 54935.34 | 531.50 ±78.46 |

in the Rb1SalB group). The OTU numbers were not significantly different among groups ($P > 0.05$).

### Overall structural changes of gut microbiota

The venn diagram showed that there were 800 core OTUs shared in all groups, accounting for 24.9%. Compared with the Con group, Rb1 and SalB reduced the unique OTUs (355 and 410), and the unique OTUs were even fewer in Rb1SalB group, only 199 specifically (Fig. 4A). The bar plots showed the relative abundance at the phylum (Figs. 4B–4C) and the genus (Figs. 4F–4G) levels among groups. Compared with the Con group, Rb1, SalB and Rb1SalB treatment significantly increased the relative abundance of Firmicutes phylum. Meanwhile, Rb1 and Rb1SalB improved the relative abundance of Bacteroidetes phylum (Fig. 4C). Furthermore, B/F ratio in Con, Rb1, SalB, Rb1SalB was 0.298, 0.492, 0.179, 0.288, respectively. The taxa with the top 35 relative abundances at phylum (Figs. 4D–4E) and

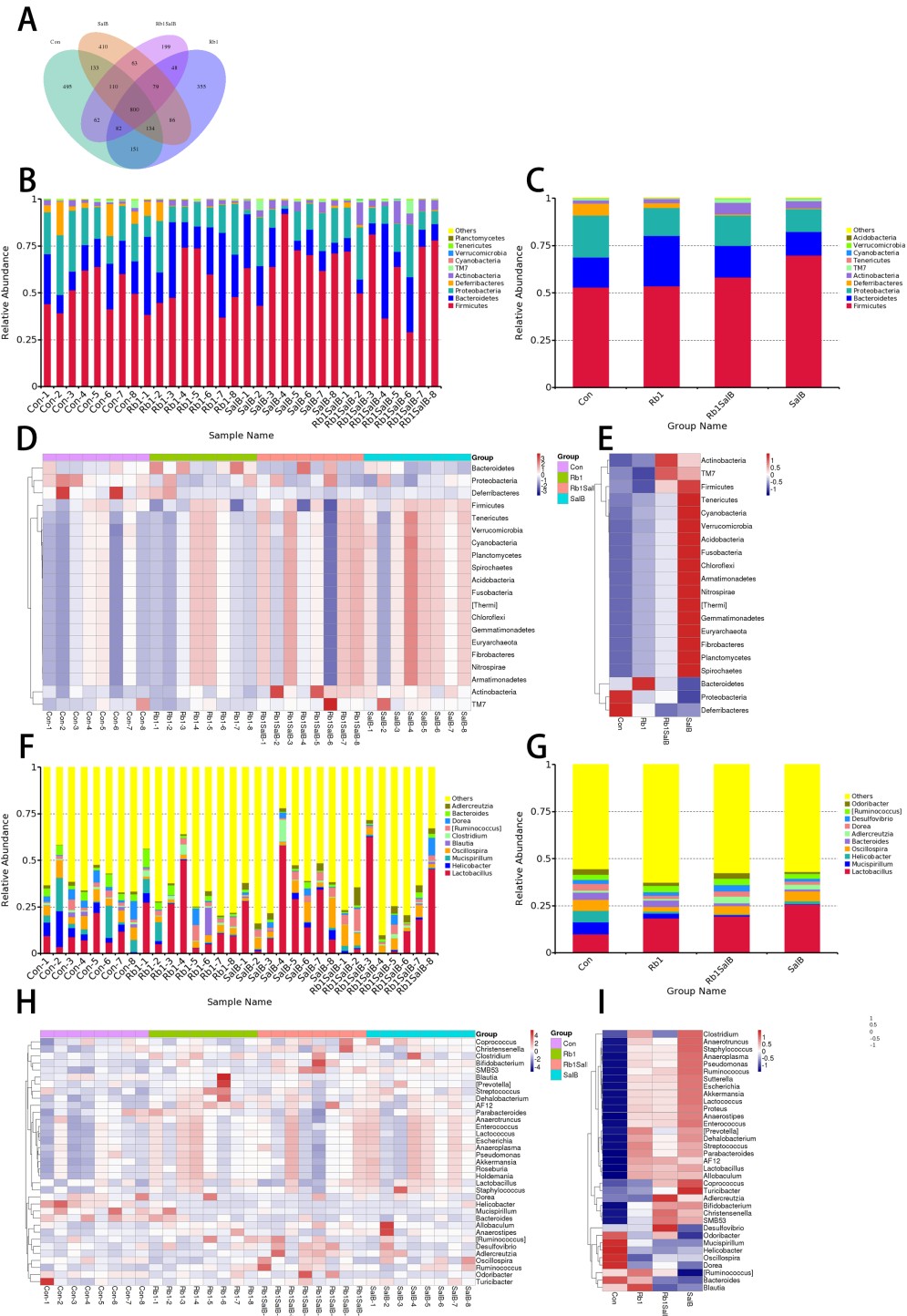

**Figure 4** **OTUs analysis and species annotation results.** (A) Venn diagram of each group. The overlapping part represents the number of shared species with other groups, while the non-overlapping part represents the number of unique species in the group. (B, C) Bar plot of relative abundance in each sample (group) at phylum levels. The different colors in the figure represent (continued on next page...)

**Figure 4 (…continued)**
different categories at phylum levels, while others refer to the relative abundance of all species except the top 10 in relative abundance. (D, E) The clustering heat map of the samples (groups) at phylum level. The colors in the heat map correspond to the Z value. Z = [RA(sample) - RA(mean)]/ SEM(RA). (F, G) Bar plot of relative abundance in each sample (group) at genus levels. (H, I) The clustering heat map of the samples (groups) at genus level. There are eight mice in each group.

genus (Figs. 4H–4I) levels were clustered into heatmaps. The results indicated the similarity and difference of taxa relative abundance among different samples, which is consistent with the results illustrated in Figs. 4B and 4F.

Next, alpha and beta diversity analysis was conducted to evaluate the diversity and overall structure of fecal microbiota, respectively. As shown in the Chao1 curve, compared with the Con ($1012 \pm 241.4$) group, Rb1 ($924.6 \pm 155.4$), SalB ($893.8 \pm 278.3$) and Rb1SalB ($742.9 \pm 110.9$) reduced the overall richness and diversity (Fig. 5A). The comparison between the Con and Rb1SalB was significant ($P < 0.05$). However, the results from Shannon index showed no significant difference among Con ($5.88 \pm 0.28$), Rb1 ($5.26 \pm 0.45$), SalB ($5.35 \pm 0.99$) and Rb1SalB ($5.16 \pm 0.78$) group (Fig. 5B, $P < 0.05$). As to beta diversity analysis, the non-metric multi-dimensional scaling (NMDS) analyses based on weighted (Fig. 5C) and unweighted unifrac distance (Fig. 5D) were conducted. Then we conducted analysis of molecular variance (AMOVA) and the results showed that apart from microbiota from SalB vs Rb1SalB, the difference between other group pairs is significant ($P < 0.05$, Table S1).

## Key indicator taxa of gut microbiota corresponding to Rb1, SalB and Rb1SalB treatment

In this study, LEfSe analysis was carried out to identify the marked indicator taxa corresponding to the treatment. Compared with the Con group, Rb1 mostly decreased Helicobacteraceae and Ruminococcaceae, and enriched Rikenellaceae at family level. Rb1 also decreased microbiota *Dorea*, *Helicobacter* and *Oscillospira* genera significantly (Figs. 6A–6B). SalB reduced Helicobacteraceae, Odoribacteraceae, Bacteroidaceae and Deferribacteraceae at family level, *Odoribacter*, *Bacteroides*, *Helicobacter*, *Mucispirillum* at genus level (Figs. 6C–6D). Rb1SalB treatment enriched Coriobacteriaceae, decreased Bacteroidaceae, Helicobacteraceae and Deferribacteraceae at family level; increased *Adlercreutzia*, reduced *Dorea*, *Bacteroides*, *Helicobacter* and *Mucispirillum* at genus level; and markedly decreased the relative abundance of two species, *Helicobacter pullorum* and *Mucispirillum schaedleri* (Figs. 6E–6F).

In order to better understand the association between metabolic indicators and annotated microbiota at phylum level, we next conducted Spearman correlation analysis (Fig. S2). Between the Con and Rb1 group, Deferribacteres was found positively correlated with FBG and HDL-c content ($P < 0.05$); and Proteobacteria was positively correlated with HDL-c level ($P < 0.05$, Fig. S2A). Between the Con and the SalB group, Actinobacteria was negatively correlated with HbA1c level ($P < 0.05$); while Deferribacteres was found positively correlated with HDL-c, FBG and HbA1c level ($P < 0.05$, Fig. S2B). Between Con and Rb1SalB group, Actinobacteria was found negatively related to HDL-c, TC, TG and

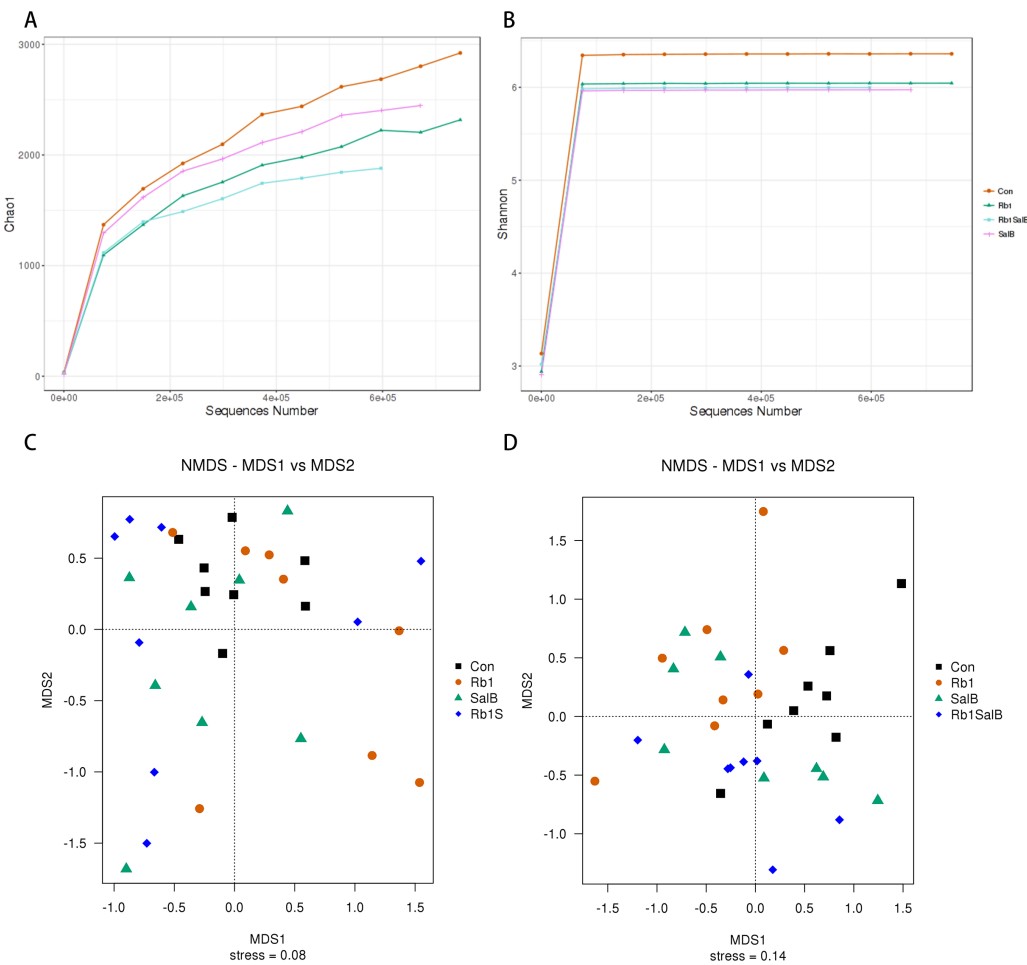

**Figure 5  Alpha and Beta diversity analysis.** (A) Chao1 curve. (B) Shannon curve. Values shown are Chao1 or Shannon index of sequences number. Legends of different colors represent different groups. (C–D): Non-metric multi-dimensional scaling analysis based on weighted unifrac distance matrix (C) and unweighted unifrac distance matrix (D).

FBG level; while Deferribacteres was positively related to HDL-c, TC, TG, FBG level and body weight gain ($P < 0.05$; Fig. S2C).

## DISCUSSION

Obesity and obesity-related diseases like type 2 diabetes are the consequences of both genetic and environmental factors, and the interaction between host metabolism and gut microbiota takes active part in the pathogenesis of these metabolic disorders (*Barlow, Yu & Mathur, 2015*; *Zhang et al., 2013*). Researchers who performed their studies about human obesity among hunter-gatherer groups claim that it is the caloric intake instead of a sedentary lifestyle that have resulted in high rates of obesity in Western countries (*Pontzer et al., 2012*). Besides high level daily activities, diets in hunter-gatherer groups usually contain less calories and are rich in fibers and micronutrients
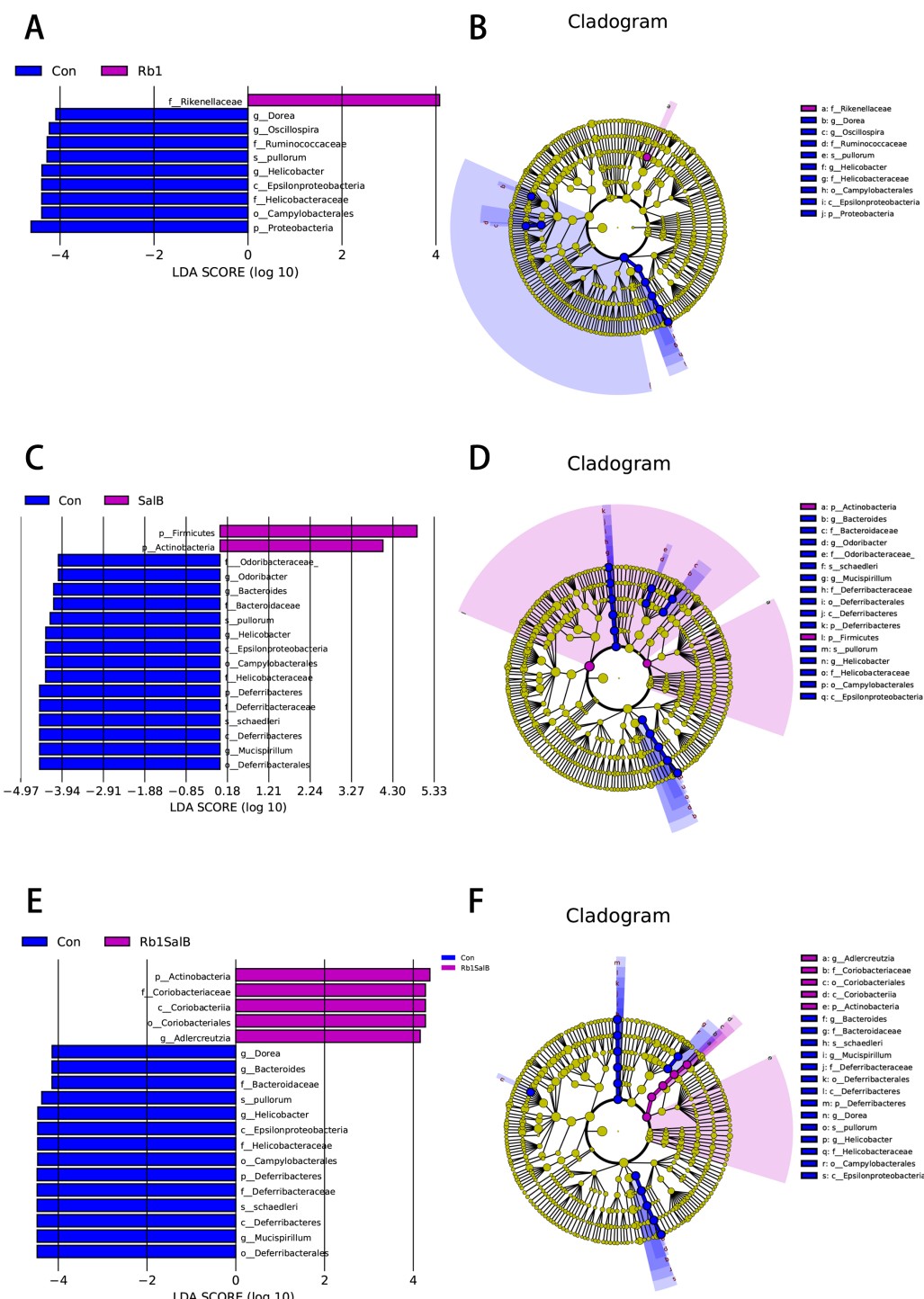

**Figure 6** **Linear Discriminate Analysis (LDA) effect size plot.** (A) LDA score and (B) cladogram of control and Rb1 group. (C) LDA score and (D) cladogram of control and SalB group. (E) LDA score and (F) cladogram of control and Rb1SalB group. The abscissa is the LDA score, and only the significantly differentiated species of which LDA score > 4 are presented. Colors of legends refer to different groups. In the cladogram, yellow nodes refer to species that are not significantly different, while red or green nodes refer to species with high LDA score in the group.

(*Pontzer, Wood & Raichlen, 2018*). This raises the potential of medicinal plants (rich in fibers, active ingridients and micronutrients) in management of metabolic diseases. Ginsenoside Rb1 and salvianolic acid B are both active ingredients from traditional herbal medicine. Ginsenoside Rb1 belongs to ginsengdiol saponins and is rich in Panax ginseng C.A.Mey. (Renshen), Radix et Rhizoma Notoginseng (Sanqi), and Panax quinquefolius L. (Xiyangshen). Recently, the pharmacological actions of ginsenoside Rb1 have been explored, and several studies have shown that ginsenoside Rb1 exerts anti-obesity and anti-diabetic effects in DIO or diabetic mice (*Park, Kim & Shim, 2019*; *Zhou et al., 2019*). Salvianolic acid B, a major polyphenolic compound from Salvia miltiorrhiza Bunge, exhibits potential ability to ameliorate glucolipotoxicity and restore metabolic homeostasis. Besides, salvianolic acid B also plays a dominant role of suppressing inflammation both in inflammatory bowel diseases as well as HFD induced low-grade inflammation (*Wang et al., 2017*; *Wen et al., 2013*). As to the safety concern about these two compounds, so far there is no record about the acute toxicity of ginsenoside Rb1. According to clinical application experience, ginsenoside Rb1 has good safety profile. *Ding et al. (2017)* examined the safety of salvianolic acid B, and by intravenous injection, they claimed that LD50 of salvianolic acid B was 646.8909 mg kg$^{-1}$. Both of the two compounds showed low toxicity and thus suitable for clinical application and further investigation. Consistent with previous studies, we observed that after 8 weeks of administration, ginsenoside Rb1, salvianolic acid B and their combination exhibited beneficial effect on reducing blood glucose level and lipid content of DIO mice. In addition, these two ingredients seemed to prevent the high fat diet induced weight gain. However, in this study we did not manage to measure food intake and daily activity of each mouse so as to the exclude the effect of energy consumption on body weight (even though our previous results already showed these two compounds did not affect food intake, Fig. S1), and these will be included in future study.

To explore the underlying mechanisms, we observed the gut microbiota composition of the mice in each group. Under the circumstances of obesity and diabetes, the diversity and composition of gut micriobiota present definite changes (*Barlow, Yu & Mathur, 2015*; *Qin et al., 2012*). Our data showed ginsenoside Rb1, salvianolic acid B and their combination reduced the overall gut microbiota diversity and richness. It appeared that these two ingredients mimic antibiotic action. However, our data is similar to some findings that taxa diversity was decreased after treatment with probiotics or prebiotics, which might be related to the dosage of medication and a comprehensive result of suppressing the harmful taxa and increasing the beneficial ones (*Jiao et al., 2019*; *Wang et al., 2019a*; *Wang et al., 2019b*). And the sampling location (feces or mucosa samples) affects the microbiota diversity as well. Moreover, we found that compared with mice in Con group, Rb1 and Rb1SalB improved, while SalB treatment decreased the relative abundance of Bacteroidetes phylum. The relative abundance of Proteobacteria phylum was lowered in all three treatment groups. Our data were not completely consistent with the studies that demonstrated the negative association of B/F ratio and BMI (*Ley et al., 2005*; *Turnbaugh et al., 2006*). However, accumulating researchers found conflicting results and the practicability of this ratio still remains to be verified (*Lee, Sears & Maruthur, 2019*). According to the study of Murphy et al., changes of phyla Bacteriodetes and Firmicutes were not relevant to energy

harvest in DIO mice (*Murphy et al., 2010*). The Bacteroidetes also include both beneficial and pathogenic taxa, and we have to make a judgment on specific condition (*Wexler, 2007*). Therefore, we focused on the significantly altered taxa to explain the mechanism of the treatment of ginsenoside Rb1, salvianolic acid B and their combination in HFD induced obese mice.

In this study, we observed significant alteration in seven families, seven genera and two species between the control and treatment groups. Firstly, at family level, Rb1 treatment enriched Rikenellaceace, and decreased Ruminococcaceae and Helicobacteraceae abundance. SalB treatment reduced Helicobacteraceae, Odoribacteraceae, Bacteroidaceae and Deferribacteraceae abundance. Meanwhile their combination also increased Coriobacteriaceae abundance. It was confirmed that HFD causes Rikenellaceace and Coriobacteriaceae reduction, and the restoration should be beneficial in anti-obesity treatment (*Alard et al., 2016*; *Clarke et al., 2013*; *Thomaz et al., 2020*). Furthermore, Coriobacteriaceae abundance is found to be positively correlated with GLP-1 secretion, the latter of which contributes to glucose homeostasis (*Cornejo-Pareja et al., 2019*). Besides, family Ruminococcaceae is found to be positively associated with long-chain polyunsaturated fatty acid (PUFA)-including TGs in postprandial response (*Bondia-Pons et al., 2014*). The abundance of this family usually increases after HFD induction and associates with impaired glucose tolerance (*Zhang et al., 2013*). Secondly, at genus level, *Dorea*, *Oscillospira* and *Helicobacter* abundance were reduced by Rb1. Besides *Odoribacter*, salvianolic acid B also depressed *Bacteroides*, *Helicobacter* and *Mucispirillum*. Furthermore, treatment of the two ingredients together enriched *Adlercreutzia* abundance. In obese rodents, increased abundance of *Dorea*, *Odoribacter* and *Oscillospira* was observed, and their metabolites, short chain fatty acids were claimed to provide extra energy to the host which might contribute to fuel surfeit (*Garcia-Mazcorro et al., 2018*; *Jiao et al., 2019*; *Jiao et al., 2018*; *Hamilton et al., 2015*). SCFA produced by these microorganisms are important energy sources for both the intestinal flora itself and intestinal epithelial cells. In addition to being a local substrate for energy production, SCFAs can also inhibit histone deacetylase activity or act as ligands for G protein-coupled receptors (GPCRs). Combining with the GPCRs, such as GPR41 and GPR43 on the surface of epithelial cells, SCFA regulate the homeostasis of host cells in the intestines through the serum peptide tyrosine-tyrosine, glucagon-like peptide 1, or peroxisome proliferator activated receptor γ (*Ottosson et al., 2018*). Moreover, the occurrence of obesity is often accompanied by subclinical inflammation (*Li et al., 2017*). It has been reported that Toll-like receptors (TLR) expressed in intestinal epithelial cells can recognize changes in intestinal microbiota and regulate inflammation through NF-κB signaling pathway in obese rodents (*Tilg et al., 2020*). In addition, *Dorea*, claimed to be positively related with BMI, was in our study decreased after treatment (*Ottosson et al., 2018*). Another genus *Adlercreutzia*, which was found to be positively associated with leanness and lowered in diabetic female patients (*Moon et al., 2018*; *Caesar et al., 2015*), was enhanced markedly after treatment with ginsenoside Rb1 and salvianolic acid B. Besides, depleted genus *Helicobacter* is represented by *Helicobacter pylori* (*Singh et al., 2015*). *H. pylori* infection is known as a risk factor of less favorable lipid files and increased BMI (*Suki et al., 2018*; *Zhao et al., 2019a*; *Zhao et al., 2019b*). Dyslipidemia and increased BMI

both contribute to the metabolic disorder, which means *H. pylori* might be associated with metabolic diseases. Finally, two significantly depressed species found in present study, *Helicobacter pullorum* and *Mucispirillum schaedleri*, are claimed to play dominant role in inflammation and considered as the possible pathogens of certain diseases (*Javed et al., 2017*; *Loy et al., 2017*).

The correlation analysis revealed the association between metabolic indicators and annotated microbiota at phylum level. Unlike previous researches, we did not find significant positive association between Bacteroidetes and metabolic indicators measured in our study. However, our results revealed two other phyla, Actinobacteria and Deferribacteres that exhibit strong relationship with glucose and lipid content of obese mice. We found positive association between Deferribacteres and FBG, TC, TG and body weight gain. Actinobacteria was found negatively related to glucose and lipid indicators such as HbA1c, FBG, TC, and TG. In a previous study, Deferribacteres was found significantly increased in T2DM compared to normal population and negatively correlated with FBG level (*Nuli et al., 2019*). Also, our results were consistent with this discovery. On the other hand, Actinobacteria play provital role in metabolism of plant-derived carbohydrate starch and polysaccharides through glycosyl hydrolases and release SCGAs (*Binda et al., 2018*). These indicated that both Deferribacteres and Actinobacteria might exert important influence on host gut homeostasis and glucolipid metabolism. Besides Bacteroidetes and Firmicutes that already attract most attention, we should also focus on other bacterial phyla and investigate their role in host metabolism.

Taken together, ginsenoside Rb1, salvianolic acid B and their combination altered gut microbiota composition in obese mice, which to be more specifically, including the decreased abundance of opportunistic pathogen and obesity related microbial communities and the increased abundance of leanness related bacteria. Based on current findings, significant taxa corresponding to our treatment are identified, some of which play important roles in microbiota metabolites like SCFAs. Further investigations with larger samples are needed to elucidate the sophisticated changes of gut microbiota and their metabolites after application of ginsenoside Rb1 and salvianolic acid B.

## CONCLUSIONS

In conclusion, ginsenoside Rb1 and salvianolic acid B protected from high fat diet induced glucolipid disorders, which might be associated with the alteration of gut microbiota. Our study indicates that ginsenoside Rb1 and salvianolic acid B are both promising drug candidate from natural herbs that exhibit therapeutic effect on obesity-related metabolic disorder partly through modulating gut microbiota composition. This paves the way for future research and provides inspiration for pharmaceutical investigation.

## ACKNOWLEDGEMENTS

We are grateful to prof. Jinxing Lu from the Chinese Center for Disease Control and Prevention who helped us revise the microbiology terms.

### Funding

This research project was funded by the National Natural Science Foundation of China (No. NSFC81503540 & NSFC81274041), the National project for leading talents of traditional Chinese Medicine–Qihuang scholar Project (No. 10400633210005), the Beijing Joint Construction Project (No. 0101216-14&0101216-2013) as well as the Key Drug Development Program (No. 2012ZX09103201-005). The funders had no role in study design, data collection and analysis, decision to publish, or preparation of the manuscript.

### Grant Disclosures

The following grant information was disclosed by the authors:
National Natural Science Foundation of China:  NSFC81503540, NSFC81274041.
National project for leading talents of traditional Chinese Medicine–Qihuang scholar Project: 10400633210005.
Beijing Joint Construction Project:  0101216-14&0101216-2013.
Key Drug Development Program: 2012ZX09103201-005.

### Competing Interests

The authors declare there are no competing interests.

### Author Contributions

- Ying Bai performed the experiments, analyzed the data, prepared figures and/or tables, authored or reviewed drafts of the paper, and approved the final draft.
- Xueli Bao, Ruyuan Zhu, Chenyue Liu and Fangfang Mo performed the experiments, authored or reviewed drafts of the paper, and approved the final draft.
- Qianqian Mu and Xin Fang analyzed the data, authored or reviewed drafts of the paper, and approved the final draft.
- Dongwei Zhang, Guangjian Jiang and Ping Li analyzed the data, prepared figures and/or tables, authored or reviewed drafts of the paper, and approved the final draft.
- Sihua Gao and Dandan Zhao conceived and designed the experiments, authored or reviewed drafts of the paper, and approved the final draft.

### Animal Ethics

The following information was supplied relating to ethical approvals (i.e., approving body and any reference numbers):

The Animal Ethics Committee of Beijing University of Chinese Medicine provided full approval for this study (NO. BUCM-4-2016061701-3001).

### Data Availability

The raw data are available in the Supplemental Files. Data used to create Figures 4–6 are available at the Sequence Read Archive (SRA): PRJNA610166.

## Supplemental Information

Supplemental information for this article can be found online at http://dx.doi.org/10.7717/peerj.10598#supplemental-information.

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
