# Peer review of "Ginsenoside Rb1, salvianolic acid B and their combination modulate gut microbiota and improve glucolipid metabolism in high-fat diet induced obese mice"

_PeerJ, doi:10.7717/peerj.10598_

## Round 0.1 · original submission · Major Revisions

The study is interesting, although the authors should improve several aspects of their manuscript prior to publication.

·

Basic reporting

L82: actually there are studies on the effects of SalB on the gut microbiome: Wu et al. (2018, Journal of Functional Food) and Li et al. (2020, Food & Function).

L164: with multi-panel figures, it is better to refer to specific panels instead of the whole figure in general following relevant descriptions in the main text.

The first part of the discussion (L234-259) contains largely background information unrelated to findings of the current work and should be removed or simplified and merged with the background section.

It would help if there was a figure showing the whole experimental design (L110).

The existing figures should also be improved as follows: in general, labels of figure panels should be placed on top of each diagram instead of under them; y-axis labels are missing in panels D and F of Fig. 2, keys in these panels are also redundant and should be removed; the five panels in Fig. 3 should be annotated as A-E as in other figures; the text in Fig. 4 and 5 is too small to read, especially that in panel C and E of Fig. 4; the cladograms in Fig. 6 are of poor quality with unreadable text (Consider removing them as the same information is already shown on the bar plots to the left); Fig. S1 is an important figure and should be made a main figure (L216).

The English language of the manuscript needs to be improved. In particular, the usage of “respectively” is incorrect throughout the manuscript (L30, Fig. 2 legend L5, Fig 4 legend L5 and L8).

Numerous scientific terms were not used accurately or precisely throughout the manuscript.
For instance, it should be “taxa” instead of “microbiome” (L273) or “phylotypes” (L287, L324) or “species” (L145 Fig. 6 legend L4), “OTUs” instead of “microbiome” (L203) or “species” (Fig. 4 legend L3-4), ‘indicator taxa’ instead of ‘phenotypes’ (L41) or “phylotypes” (L219) or “marked species” (L220), “distance matrices” instead of “measurement indicators” (L153), “diversity and overall structure, respectively” instead of “richness” (L212), “taxa at different taxonomic levels” instead of “species or microbial communities” (L221), etc. Also, scientific names and genus names should be in italics throughout the manuscript (e.g. L74-75, L224). The revised manuscript should be checked by a microbiologist or scientist with relevant background.

All abbreviations should be defined at their first use in the main text (E.g. L125). Please carefully check for the rest throughout the manuscript.

References are missing for some softwares/methods used, e.g. QIIME (L149), UniFrac (L152) and LEfSe (L156).

Add a sentence somewhere in the manuscript stating that the raw sequence data is available in NCBI Sequence Read Archive (SRA) under the accession PRJNA610166.

Experimental design

Methods of the molecular analysis (L131) are not described in sufficient detail. For instance, which kit was used in DNA extraction (L135)? Which region of the 16S rRNA gene was amplified (L135)? With which primers and what recipes and conditions? How were the PCR products purified (L136)? How was the library quality controlled (L136) and checked (L137)? Was the paired-end sequencing a 2 x 300 bp one (L137)? How were the paired-end reads assembled/merged and quality filtered (L138)? How were the chimeric sequences checked and removed (L139)? Which reference database was used in taxonomic identification (L142)? Which version of QIIME was used (L149)? Was the NMDS analysis conducted using QIIME as well (L155)?

Other method-related queries include: How many mice were there per cage (L91)? Were both tap water and food sterilised (L92)? With standard chow (L112)? Why these concentrations (L117-118)? Information of the kit used for HbA1c detection is lacking in the Reagents and equipment section (L129).

L25: change the section heading to ‘Background’ and add the incentive of the study or research gap

Validity of the findings

L37: according to Fig. 1A, SalB did not decrease the body weight significantly.

L44-45: this conclusion is not supported by the data (Fig. 1A and 2A). Also on L261-262.

L45: tone down the claim, e.g. by using “may be” instead of “is”. Also on L333-334.

Without inclusion of a naive control group for comparison, it is invalid to claim that the model group has “impaired” glucose tolerance (L179-180) or “dyslipidemia” (L188-189), or that a certain treatment could “reverse” an alternation of the microbiome (L285-286).

L182-183: however, there were significant differences in the baseline values between some treatment groups and the control group (Fig. 2E).

L198-201: all sequence datasets need to be rarefied to the same (smallest) number of reads before meaningful comparisons can be made. Otherwise, it is just normal for the Rb1SalB group which has the lowest no. of reads to have the lowest no. of OTUs!

L216: did you test this statistically with e.g. PERMANOVA? It seems that the 2nd panel has good potential to show significant groupings after the test.

Additional comments

L27: mention OGTT as well

L34: provide full names of the abbreviations here

L50: the ’2’ at the end of the sentence should be in superscript

L105: incomplete sentence

L131: “16S rDNA”

L132: ‘administration of drugs’?

L145: ‘bar plots’ instead of ‘accumulative histograms’

L148: “Alpha diversity analysis”

L149: “rarefaction curves” is the preferred term

L160: “Fisher’s LSD test”

L166: ANOVA P > 0.05?

L169: it seems that Rb1 also showed significant difference in Fig. 1A. Mention it here as well

L190: TC as well

L191: add “Fig. 3” here

L192: add “P > 0.05”

L195: “4,002,927”

L198: “OTUs”

L205: remove the extra “b”

L207: were the increase statistically significant?

L208: the decrease was too trivial to mention at all

L210-211: I think this is not these figures meant to show!

L214: support the statement with statistical test. Also, “richness and diversity”

L215: it is unclear which panel of Fig. S1 refers to which matrix

L219: “Rb1SalB”

L220: define LEfSe at first use (L156) instead

L224: “genera” instead of “genus”

L227: “Bacteroides”

L228: “Coriobacteriaceae” and “Deferribacteraceae”

L230: “Helicobacter pullorum and Mucispirillum schaedleri”. Also on L318

L267-268: provide a reference for the statement

L284-285: sentence of unclear meaning

L289: add “between the control and treatment groups”

L293: “Coriobacteriaceae”

L295: what kind of adjustment? increase or decrease?

L296: is the increase a good one or a bad one in this context?

L302: Helicobacter as well

L303: “enriched” instead of “enhanced”. Also on L313

L314: “depleted” instead of “down regulated”

L315: what do you mean by this sentence? The year information is missing here. Also, “H. pylori” instead of “HF”

Fig. 1 legend L4: remove “change”

Fig. 2 legend L5: why only 4 out of 8 mice were selected for this analysis?

Fig. 4 legend L9: does “RA” mean relative abundance?

Fig. 6 legend L1: “Linear”

Table 1: the number of decimal places used should be consistent

Reviewer 2 ·

Basic reporting

The authors of the article titled ‘Ginsenoside Rb1, salvianolic acid B and their combination modulate gut microbiota and improve glucolipid metabolism in high-fat diet induced obese mice’ describes the effect of herbal extracts on metabolism through modulation of gut microbiota in obese mice model. The manuscript is well written in terms of understanding the finding, however it needs significant improvement before acceptance.
ALL the figure quality needs to be significantly improved in order to make the texts inside readable. In figure 1, reduce line thickness and clear indication of the comparing groups for statistical significance is lacking.
I have included the language edits as minor comments:

Line 49: the starting sentence in the 'introduction' needs to be changed or removed because it reads inappropriate.
Line 56: mention some of the ‘short and long term medical consequences’.
Line 59: change ‘emerged’ to ‘increased understanding’ because it was always there, we just identified its role in health and disease
Line 61: cite a reference discussing contribution of microbiota to human biology.
Line 62: change ‘closely related’ to ‘influenced by’
Line 63: change ‘dysbiosis is essential in’ to dysbiosis contributes to’
Line 67: it is confusing to understand here, ‘their’ means whose homeostasis, should be the host but it reads like it is of the microbiota.
Line 71: change ‘proportion’ to relative proportion
Line 72: whose metabolites? The authors can break the sentence here and explain the context of the metabolites from the microbiota or from the host itself.
Line 76: needs a reference to be cited, describing role of ginsenoside and salvianolic acid on various diseases.
Line 130: provide the kit details, e.g. catalogue number, vendor etc.
Line 189: cite reference for the statement that ‘long-term HFD feeding caused dyslipidemia’
Line 198: typo ‘OUTs’ should be OTUs
Line 293: reference for Coriobacteriaceae should be included. The references cited here only talks about Rikenellaceae
Line 325: remove ‘this inspired us that’
Lastly, in the discussion section there is inappropriate use of the word ‘meanwhile’ several times in the discussion. Authors should use their vocabulary to replace this word.

Experimental design

Major improvement in the text includes the following:

Line 51-54: discuss in brief about the hunter-gatherer groups where physical fitness was necessary to gather food and cite appropriate references. Review recent papers of Janelle Ayres.
The authors should either experimentally show that relative abundance of Firmicutes and Bacteriodetes controls obesity OR, discuss extensively several studies that supports this phenomenon as mentioned in section 4.2.
In the results section, a concluding statement should favorably be included.
In the discussion, authors should also include possibility of the direct effect of these products on host cell metabolism apart from changing the microbiota. If the receptors are identified or mention a speculated mode of entry into the cells. Moreover, the last paragraph has a lot of repetition and can be significantly reduced. Names like ‘Dorea’ has been mentioned several times which is unnecessary.
The authors should include a pathway/signaling description about how the glucolipid metabolism changes on administering these extracts to support their argument about reducing obesity.
I have no comments on the conclusion section.

Validity of the findings

The authors should mention the number of times they repeated the experiments to support their observation. Statistical significance should be included with +/- SD for all the experiments, as the statistical significance is described for the sequence analysis.

Additional comments

This study is important in terms of evaluating the potential of herbal compounds for treatment of ever-increasing metabolic disorders around the globe. It also highlights the role of gut microbiota composition and relative abundance in controlling metabolism in mammals. To increase visibility of this study and relate multiple aspects of biology, the manuscript needs to address most of the comments before being considered acceptable for publication. I appreciate the authors’ intention to evaluate therapeutic potential of herbal products for the treatment of metabolic disease e.g. obesity. However, I think addressing the above comments and including them into the text would significantly improve its validation, quality and would be appealing to a greater audience.

·

Basic reporting

The authors show the role of ginsenoside and salvianolic acid in glucolipid metabolism. A big portion of the study is repetition of previous work. The figures are relevant but need some modifications. The experiments are mostly well executed with control; the exceptions are mentioned in general comments.

Experimental design

The study is of importance. A major portion is repetition of previous work. Since, PeerJ accepts replication experiments, this study comes within the scope of PeerJ. The research plan is well defined but further validation experiments need to be done to link the change of gut microbiota to glucolipid metabolism.

Validity of the findings

The study looks statistically sound. Conclusions are well supported by results except the ones that I have mentioned in the general comments.

Additional comments

Overall, the study is interesting. Although, there are certain topics that need attention as followed.

1. The authors wrote in the abstract (line 81-82) that the effect of Rb1 and SalB in regulating gut microbiota is not known but there are studies which looked at it previously. For example, check the review by X. An et al., 2019 in Biomedicine & Pharmacotherapy. I would suggest correcting that statement accordingly and cite proper references.
2. Fig 1, 2, 3 and to some extent Fig 4 is repetition (Ref. S. Lee et al.,Int J. Mol. Sci. 2012 and D. Zhao et al., JTCMS, 2017 etc.) of previous work and there is very less new information.
3. In the first half, for the body weight and blood glucose studies, an essential experiment is missing; the effect of these plant products (Rb1 and SalB) in food intake and activity of mice is not followed. Since weight gain and blood glucose can be affected by change in food habit or physical activity, these aspects cannot be overlooked.
The latter half of the study shows the effect of Rb1 and SalB on gut microbiota alteration and they relate this change to the metabolic effect they saw. Feeding mice with any plant product might change the gut microbiota but linking that change to some metabolic effect need further validation, which is missing in this study! In the current form, the authors only show change in the gut microbiota on feeding Rb1 and SalB and tried to draw a correlation with metabolism, but it is inconclusive and need further experimental results.
4. The writing in the ‘results’ section is weak and needs to be improved.
5. For Fig 1B, the Y-axis label could be replaced with ‘% body weight change’. The current label is confusing to a reader. Also, the ‘*’ used to denote level of significance are randomly placed; it should ideally be placed at the top of the data points.
6. For Fig 2: The Y- axis label for 2D, F is missing.
7. For Fig 4: Figure labels are small, unreadable, and pixelated. I would suggest replacing with better quality picture. Same is true for Fig 6 cladograms.

---

## Round 0.2 · Minor Revisions

The authors have substantially improved their manuscript. However, based on the reviewers' comments, there are still some minor issues to be addressed prior to publication.

·

Basic reporting

L98: somehow the given name “Xinyue” is incorrectly used as the surname in the citation of “Wu et al. (2018)” here. A quick look also identified some other similar mistakes, e.g. it should be “Pan et al. 2018” instead of “Yanyun et al. 2018” on L604. Please carefully check all of the citations again.

L185: the correct citation for QIIME should be Caporaso et al. (2010, nature methods)

L381: “H. pylori” in italics

Figure 1: a legend for the newly added panel A is missing

Figure 3: mention the panel numbers in suitable places of the legend

Figure 4: the legend needs to be updated with the newly named panels. Also for Figure 6.

Figure 6: “Linear” is still misspelt in the legend

Experimental design

L161: is it the “TIANamp Stool DNA Kit”? Name it explicitly

L163: provide proper citation(s) for the primers

L167: it is still unclear what purification kit was used along with the beads

L171: was the “2 x 100 bp” or “2 x 125 bp” chemistry used?

L177-178: it is still unclear which “reference database” was used in assigning taxonomy to the reads

Validity of the findings

L47-48: According to results of Figure 1B and 2A, I am still not convinced by the claim that “the combination of ginsenoside Rb1 and salvianolic acid B showed a better effect on decreasing body weight and improving glucose tolerance”. Also for L313-314.

L264-265: given that statistically significant grouping is now detected, this sentence should be removed

L268: is this analysis based on the weighted or unweighted unifrac distance?

Reviewer 2 ·

Basic reporting

The authors have addressed all the comments on a point-by-pont manner which has significantly improved the manuscript quality.

Experimental design

Experimental observations are aligned with evaluation and discussion of the finding.

Validity of the findings

The authors must include the limitations of using these drugs, if being readily brought into applications and describe the potential side effects as a possibility.

Additional comments

I believe the minor comments are addressed meticulously by the authors including grammatical errors and typos which I overlooked.

·

Basic reporting

Several improvements are made in the revised manuscript.

Experimental design

Well done.

Validity of the findings

Acceptable

Additional comments

Should go through and correct the typos and other mistakes, such as in Line 297- ‘that have results in…’- should be - have resulted in.

---

## Round 0.3 · Minor Revisions

Please fix fig. 6 as indicated by the reviewer 1.

·

Basic reporting

no comment

Experimental design

no comment

Validity of the findings

no comment

Additional comments

Thank you for the revisions.
However, there is one more point to fix:

Figure 6: The panel numbers in the legend do not align with those shown on the figure.

Reviewer 2 ·

Basic reporting

I reviewed the answer to the comments and they are acceptable.

Experimental design

Experimental design is aligned to the question addressed.

Validity of the findings

Based on evidence provided the authors claim seems to be justified.

Additional comments

The authors have addressed all comments and the manuscript is suitable for acceptance.

---

## Round 0.4 · accepted · Accept

The authors have addressed all the issues raised by the reviewers.